# Diversity and Phylogeny of Novel Cord-Forming Fungi from Borneo

**DOI:** 10.3390/microorganisms10020239

**Published:** 2022-01-22

**Authors:** Rachel Foster, Hanna Hartikainen, Andie Hall, David Bass

**Affiliations:** 1Department of Life Sciences, Natural History Museum, Cromwell Road, London SW7 5BD, UK; Hanna.Hartikainen@nottingham.ac.uk (H.H.); david.bass@cefas.co.uk (D.B.); 2School of Life Sciences, University of Nottingham, Nottingham NG7 2RD, UK; 3Core Research Laboratories, Natural History Museum, Cromwell Road, London SW7 5BD, UK; a.hall@nhm.ac.uk; 4Centre for Environment, Fisheries and Aquaculture Science (Cefas), Weymouth DT4 8UB, UK

**Keywords:** Agaricomycetes, basidiomycetes, ITS1, ITS2, 5.8S, mycelial cords, rhizomophs

## Abstract

Cord-forming (CF) fungi are found worldwide; however, tropical CF fungi are poorly documented. They play an essential role in forest ecosystems by interconnecting nutrient resources and aiding in the decomposition of plant matter and woody litter. CF fungi samples were collected from two forest conservation sites in the Sabah region of Malaysian Borneo. Sequencing and phylogenetic analysis of the ribosomal rRNA gene array 18S to 28S region from cords collected placed all of the collected specimens in Agaricomycetes (Basidiomycetes), specifically within the orders Trechisporales, Phallales, Hymenochaetales, Polyporales, and Agaricales. Comparison of the cord-derived sequences against GenBank and UNITE sequence databases, as well as phylogenetic analyses, revealed they were all novel sequences types. Many of these novel lineages were found to be closely related to other basidiomycetes commonly found in tropical forests, suggesting a large undiscovered tropical fungal diversity in Borneo that has been detected independently of sampling fruiting bodies. We show how these sequence types relate to the morphologies of the cords from which they were sampled. We also highlight how rapid, small-scale sampling can be a useful tool as an easy and relatively unbiased way of collecting data on cord-forming fungi in difficult-to-access, complex forest environments, independently of locating and sampling sporophores.

## 1. Introduction

Some saprotrophic fungi are able to grow mycelial cords formed by aggregations of aligned hyphae, which interconnect nutrient resources, allowing them to translocate carbon, nitrogen, and phosphorous over several metres [1,2,3]. These mycelial cords are usually found forming networks between woody litter components on the floor of boreal, temperate, and tropical forests, as well as in tropical canopies, allowing fungi to forage for new resources [4]. Cord-forming (CF) fungi contribute to nutrient cycling through the decomposition of wood and plant matter, and larger, more extensive fungal systems can act as long-term nutrient sinks, allowing them to buffer against nutrient loss from the forest ecosystem [5,6]. 

Mycelial cord-like structures, known as rhizomorphs, can also be formed by ectomycorrhizal (ECM) fungi, also giving them the ability to explore for resources and translocate nutrients [7,8]. ECM mycelia are also important in the mobilisation of nutrients from both organic and inorganic sources for redistribution in the forest ecosystem and the delivery of carbon below ground, both in competition and in cooperation with cord-forming species of saprotrophic fungi [9].

Saprotrophic fungi are known to enhance biodiversity by creating different habitat patches and are also a food resource, benefiting diverse communities of insects, mosses, and other fungi [10]. Despite their important role in the forest ecosystem, the diversity of CF fungi in the tropics is not well characterised. It is hypothesised that a greater diversity of microhabitats and microclimates are present in tropical forests compared to temperate forests, which promotes the increased diversity of species. In addition, some ecological niches are completely absent in temperate forests, such as litter-trapping fungi, which are found in tropical forest canopies [11]. Tropical forest ecosystems have diverse basidiomycete communities of saprotrophic ephemeral wood-decomposing fungi on the forest floor [12]. The decomposer community is highly competitive in tropical forests due to nutrient-poor, highly weathered soils [13], and decomposers are stratified with a preference for fresh leaf litter, course or fine woody debris, or residual organic matter in the soil [2]. 

Most studies identifying fungi in tropical forests have been based on fruiting body morphology. These have shown that freshly fallen leaf litter is dominated by *Marasmius*, *Marasmiellus*, and *Mycena*. Some *Marasmius* spp. form networks in moist, tropical tree canopies, allowing them to trap and colonise falling leaf litter before it reaches the forest floor [4]. The lower and older layers of leaf litter are dominated by species of *Clitocybe*, *Collybia*, *Geastrum*, *Lepiota*, and *Leucocoprinus* [12]. Species of *Phallus* and *Megacollybia* often form conspicuous white cords, easily visible in partially degraded leaves [14]. More generally, the understory of tropical forests has a high humidity which allows saprotrophic communities to flourish, including species of *Trechispora*, *Leptosporomyces*, *Caripia*, *Deflexula*, *Fistulina*, *Polyporus*, *Pleurotus*, and *Porodisculus* [12]. These fungal communities play an important ecological role in fungal succession by binding and partially decomposing freshly fallen leaf litter reducing the loss of litter and organic soil matter through erosion [15].

Saprotrophic cord-forming fungi diversity is under-studied, both in temperate and tropical forests. In this study, we sampled vegetative fungal life stages in two tropical forest conservation areas in the Malaysian part of Borneo, largely populated by ancient tropical rainforests dominated by dipterocarp trees, which are a major source of commercial timber [16]. Dipterocarps have been shown to be associated with a wide diversity of fungal species, both ECM and saprotrophic [17], and dipterocarp diversity peaks in Borneo, where 276 species have been described [18]. Tropical rainforests, such as those found in Borneo, are rapidly being cleared for agriculture or logged for timber, an issue of global conservational concern [19]. These anthropogenic pressures have already been shown to drastically impact soil fungal communities in Malaysian rainforests, which, in turn, have the potential to impact whole forest ecosystem nutrient and energy processes [20,21,22]. 

Tropical forests such as those we sampled from in this study in Borneo can be difficult to access and are therefore under-explored. It is widely recognised there are likely to be many undescribed basidiomycete species in Borneo [23,24,25]. Mycological research specific to Borneo has been sparse since the expeditions of E.J.H. Corner in the 1930s and 1940s collecting tropical fungi species [26]. Several mycorrhizal basidiomycetes have been found to be particularly abundant in the humid tropics, suggesting these make up a rich part of fungal diversity. For example, Corner himself described 130 species of *Boletus* alone in Malaysian Borneo [23,27]. He also reported 121 species (103 of these new at the time) of *Marasmius* [25]. Molecular phylogenetics now places these *Marasmius* spp. across five genera [28], a reflection of the huge fungal diversity that is present in South East Asia. More than 30 polypore species from Malaysia described by Corner are known only from their type localities, suggesting that Malaysia is home to unique fungal communities [29]. We have been unable to find any previous surveys solely sampling vegetative (mycelial) life-stages of fungi of Malaysia in recent years. However there have been some studies of ECM diversity specific to Malaysia [17,18,21,30,31], as well as sporocarp surveys [32,33].

In this study, we used molecular techniques to identify fungal cord samples collected on an expedition to Borneo as well as gaining more insight into the evolutionary relationships of tropical CF fungi. We also showed the utility of fungal cord sampling as a rapid way of gaining insight into the diversity of active cord formers and focused on sampling different-looking cord structures to increase the possibility of novel sequence types collected with small-scale sampling effort. Sampling of the cord structures, which are very easily found and sampled, even in difficult terrain, offers an alternative approach to surveys based on sampling fruiting bodies, which may be ephemeral or appear on periodically or under particular conditions. Although fruiting bodies are useful diagnostic tools for morphological identification, they are not always useful for ecological studies because they do not accurately represent the full fungal diversity present or species abundances [34]. Furthermore, sporocarp sampling has been shown to not accurately reflect all of the species present in wood-inhabiting fungal communities, and mycelial isolation and molecular techniques can reveal additional taxa [35].

Sequencing and phylogenetic analysis of cords collected in the two forest conservation areas of Borneo sampled for this study revealed novel sequences, suggesting new, undescribed species. We also show in this study that despite the small-scale sampling with only two hours at each location, useful data were still collected. This demonstrates the value of rapid, small-scale sampling in remote regions where cost may be prohibitive for lengthy or repeated sampling visits. These are challenges often faced by many tropical regions and likely contribute to the lack of mycological study in many tropical parts of the world.

## 2. Materials and Methods

### 2.1. Sampling

Fieldwork for this study was conducted between September and October 2012 and was carried out in two locations of the Sabah region of Malaysian Borneo which are around 95 km distance from each other, both of which are protected conservation sites. The first site was Danum Valley, a conservation area of lowland dipterocarp forest in the north-eastern tip of the island (4.9618° N, 117.6892° E). The second site was the Maliau Basin conservation area, more central to the Sabah region (4.8531° N, 116.8439° E). The two sampling areas were each sampled for two hours, walking within an area of approximately two hectares, with a focus on collecting cord structures of varying morphologies to optimise collection time. Fungi were photographed in situ (next to 1.5mL Eppendorf tubes for size reference), and approximately 0.8-1cm piece of mycelial cord was collected and submerged into 1mL RNAlater (Qiagen, Hilden, Germany) in 1.5mL Eppendorf tubes. This preservation method was shown as suitable in [36]. The samples were stored at ambient temperature until DNA extraction. The cord samples were grouped on the basis of substrate, either leaf litter or tree branches, and within these substrates grouped as far as possible on the thickness/branching pattern of the gross cord morphology and whether hyphal sheets were observed.

### 2.2. DNA Extraction/PCR/Sequencing

CF fungi samples were rinsed with sterile water to remove RNALater. DNA was extracted using either the Qiagen Blood and Tissue kit, following the manufacturers protocol (lysis step was conducted overnight), or the PowerSoil DNA Isolation Kit (MoBio Laboratories, Carlsbad, CA, USA), following the manufacturer’s protocol, using a Precellys 24 homogeniser (Bertin Technologies, France) for the initial mechanical lysis step.

An 850–950 bp region of the rRNA gene array was amplified, from the 3′ end of the 18S gene to the 5′ end of the 28S (including c. 300 28S positions), therefore covering both Internal Transcribed Spacer (ITS) regions and the 5.8S gene. The forward primer ITS1 (5′-TCCGTAGGTGAACCTGCGG-3′) and reverse primer LR21 (5′-ACTTCAAGCGTTTCCCTTT-3′) from [7] were used with the following cycling conditions: 95 °C for 3 min followed by 30 cycles of 95 °C for 30 s, 55 °C for 30 s, and 72 °C for 1 min, with the final step of 72 °C for 10 min. 

PCRs were conducted in 20 μL final volumes with 1 μL of template DNA and a final concentration of 0.5 μM of each primer, 0.4 mM dNTPs, 2.5 mM of MgCl_2_, 0.2 mg bovine serum albumin (BSA), 1× Promega Green Buffer, and 0.5 U of Promega GoTaq. Fragments were visualised on 1.5% agarose gels stained with GelRed. Amplicons were Sanger sequenced at the NHM sequencing facility in both directions. Sequences were trimmed using Geneious (version 8.1). The sequences were deposited in GenBank (accession numbers: MN102414–39).

### 2.3. Phylogenetic Analysis

CF fungi sequences were blasted (using Blastn function) against NCBI GenBank nr_nt database and the UNITE database [37], and the top 3 sequences of the closest fully overlapping hits from each database were downloaded (see Appendix A). If this was an environmental sequence, the closest taxonomically characterised sequence was also retrieved. From both databases, these were aligned with sequences generated in this study using Mafft version 7, G-ins-i algorithm [38]. Hypervariable regions where alignment was ambiguous were removed using Trimal [39], with user-defined settings minimum percentage conserved positions 70%, gap threshold 0.5, similarity threshold 0.1, and window size 1. Duplicate sequences were removed from the alignment, as were any reference sequences whose length covered <60% of the alignment length. Some sequences other than the closest database matches were retained in the alignment for greater context. The resulting alignment was used to construct a Bayesian consensus tree with MrBayes v.3.2.5 [40] and a maximum likelihood rapid bootstrapped tree with RAxML BlackBox v. 8.2.12 [41]. The Bayesian analysis using two separate Metropolis coupling (MC^3^) runs with randomly generated starting trees were carried out for 4 million generations each with 1 cold and 3 heated chains. The evolutionary model applied a generalised time-reversible (GTR) substitution matrix, a 4 category autocorrelated gamma correction, and the covarion model. All parameters were estimated from the data. The trees were sampled every 1000 generations and the first 1 million generations discarded as burn-in. All phylogenetic analyses were carried out on the Cipres server [42].

## 3. Results

Twenty-seven CF fungi samples were collected from two conservation areas in Malaysian Borneo: Danum Valley and Maliau Basin (see the Materials and Methods section). In all but one case, the cords were not associated with obvious fruiting bodies, and therefore taxonomic identification could not be made on fruiting body appearance. Figure 1 and Figure 2 show the cord samples grouped on the basis of substrate, either leaf litter or tree branches, and within these substrates grouped as far as possible on the thickness/branching pattern of gross cord morphology. Sixteen of the sampled cords were growing on leaf litter (Figure 1), with cords spanning leaves, soil, and small woody debris. Nine of the sampled cords were growing directly on tree branches (Figure 2). A further single cord (BOR204, not pictured) was sampled, growing on leaves in the understory of a tree canopy.

We generated 26 18–28S sequences from 27 samples; we were unable to amplify this region for BOR99. Phylogenetic analyses (Figure 3) showed that these resolved into 20 clearly distinct lineages, all grouping within Agaricomycetes (Phylum Basidomycota). The majority of sequences branched within the orders Agaricales and Polyporales, with others in Trechisporales, Phallales, and Hymenochaetales. Figure 3 also indicates the substrate on which the fungal cords were growing and whether the samples were collected in the Danum Valley or Maliau Basin.

Some of the BOR sequences were similar to sequences on GenBank or UNITE, but none were exactly the same where comparison between them was possible (not all sequences aligned along their total length). The BOR sequences most similar to a database sequence were BOR88 and 102, which were almost identical to a sequence labelled Uncultured Agricales from UNITE (UDB024521), sampled from Laos. BOR110 was very similar but clearly not identical to UDB013330 (*Trametes* sp.) from Malaysia. The BOR85 sequence was almost identical to UDB033876 (*Psathyrella* sp.), also from Laos, but was only represented by its more conserved regions and therefore not directly comparable with the database sequence. All other BOR sequences shown as closely related to database sequences on the tree were clearly distinct from them in their variable regions when inspected on the alignment (BOR90, 98, 203, 96, 204).

Some BOR sequences were identical to each other, or differed at so few sites that they most likely represent the same lineage/species, the sequence differences being likely due to PCR/sequencing errors or intragenomic polymorphism. These are indicated on Figure 3 by vertical red bars to the right of the branch labels. A possible exception to this is BOR201, BOR78, and 79, which differed in only three positions of the aligned regions and cannot confidently be assigned to the same or different sequence types. These three samples, branching within Trechisporales, were all sampled from leaf litter substrate at Danum Valley and were sheet- and cord-forming (Figure 1a,f,o).

The lineage represented by BOR 91, 97, and 100 was also sampled three times, all from in the Maliau Basin, from leaf and branch substrates. Figure 1 shows they all formed thick white cords similar in gross morphology. They grouped in Polyporales, with the closest named sequence match in GenBank being to *Skeletocutis delicata* (MF685355), and in UNITE, *Skeletocutis brevispora* (UBD0799092). BOR110, BOR77, BOR90, and BOR82 also grouped within the order Polyporales. 

The clade marked A (Figure 3), within Agaricales, comprised mostly novel sequences generated in this study: eight sequences, representing six distinct lineages from samples from both regions, showing variable morphology and with different growth substrates (Figure 1 and Figure 3). BOR102 and 88 was noted above as likely identical to an uncharacterised Agricales sequence from Laos. *Gerronema* sp. (UDB039651), also from Laos, was similar but not identical to BOR98, and *Stereopsis* sp. (UDB033468) from Laos, was found to be close to BOR203. Other BOR sequences in this clade were mutually related but were less similar to GenBank and UNITE reference sequences than those listed above.

BOR85, BOR81, and BOR94 also grouped within Agaricales. BOR85 had a 99% ITS similarity to *Psathyrella* sp. (UDB033876, from Laos). The closest matches of BOR81 and BOR94 were *Cystodermella granulosa* (98%) and *Cortinarius delibutus* (84%), respectively. Within Agaricales, BOR204 and BOR96 branched in the clade representing the family Marasmiaceae. These two sequences were closely related to sequences from samples morphologically identified as *Marasmius*. BOR101 grouped within Phallales, with closest matches to several *Phallus* spp. BOR205 had a closest match (85%) to *Hyphodontia* sp. (Order: Hymenochaetales) collected from Cameroon.

## 4. Discussion

All of the fungi collected on this fieldwork expedition produced sequences that were not present in GenBank or UNITE sequence databases; this is unsurprising, given that fungal diversity within tropical regions is considered to be largely undiscovered [11]. All of the fungi collected were Agaricomycetes, a phylogenetically diverse group of fungi [43]. The genera represented in the tree—*Trechispora*, *Phallus*, *Ceriporia*, *Phanaerochaete*, *Odonticium*, *Stropharia*, *Psathyrella*, *Cortinarius*, *Gerronema*, *Marasmius*, and *Inflatostereum*—are known saprotrophs, and include known cord- and net-forming fungi species.

Three samples were grouped within Trechisporales, with the closest matches to the genus *Trechispora* (Figure 3), which has a widespread distribution [44], with most species being white rot wood fungi or soil-inhabiting saprotrophs [45]. Sequences BOR78, BOR79, and BOR201 likely represent the same fungal species; sequence differences between them may be attributable to sequencing/PCR errors and/or intragenomic polymorphism between multiple copies of the rRNA gene array. All three specimens had thin white cords; however, BOR78 formed sheets (Figure 1a) and BOR201 also had thicker cords (Figure 1n). If these are the same species, these differences could be explained by the age of the cord system or the growth substrate. *Trechispora* spp. are known to form sheets and cords with a gross morphology similar to the collected samples [46].

BOR77, BOR82, BOR90, and BOR110 grouped within Polyporales, a large order with around 1800 described species. Polyporales have a varied morphology ranging from bracket fungi to resupinate and corticoid (crust) forms and are known to dominate the ecological niche of wood decay in terrestrial systems [45]. The fungi collected were all cords, with the exception of BOR110, which had small fruiting bodies (Figure 1n). Within Polyporales, BOR91, BOR97, and BOR100 (almost certainly representing the same taxon) grouped as sister to *Skeletocutis brevispora* and *Skeletocutis delicatata*, two white rot polypores described in temperate forests in Estonia [47]. Another GenBank match (not shown in Figure 3, see Appendix A) was *Skeletocutis bambusicola*, originally described on decaying bamboo from the tropical region of Yunnan in China [48]. All three specimens had thick white cords, which spanned a large area, e.g., BOR100 was growing on a tree trunk and roughly spanned an area of one square meter; BOR97 and BOR100 were both found growing on wood; and BOR91 had thick white cords and therefore was grouped with thick threads, but was growing in close proximity to dead wood also (see Figure 1m,w,x).

Agaricales contains over 13,000 species, the vast majority of which are saprotrophic or ectomycorrhizal [45]. Thirteen of the 26 samples sequenced branched within Agaricales, the majority (eight) of these sequences belonging to Clade A and the larger clade within which it branches. This clade includes a diversity of closely related but apparently distinct lineages collected during this study. The closest GenBank matches are *Gerronema* spp. and *Megacollybia* spp. (not all shown on Figure 3, please see Appendix A), both considered *incertae sedis* within the Agaricales order [49]. The new sequences within the robustly supported Clade A could potentially represent novel species of *Gerronema*, a genus of lignicolous fungi with a largely tropical distribution.

Additionally grouping in Agaricales was BOR94, which had the closest match of *Cortinarius delibutus*, a known ECM fungus [50]. Figure 1d shows BOR94 in close association with soil and roots, which suggests BOR94 may also be of ectomycorrhizal ecology. BOR205 also had a close match of ECM origin, ‘JX316513 Uncultured fungus’ in Genbank (see Appendix A). This sequence was collected in a study assessing ECM fungal communities in Nothofagus forests in Patagonia, Argentina [51]. Figure 3 places BOR205 robustly in the order Hymenochaetales. The Hymenochaetales are considered to be a clade of wood-decaying species, the majority causing white rot, and some species within this order are known to form mycorrhiza [52].

Within the family Marasmiaceae, *Marasmius* is a known genus of cord-forming fungi; some tropical species form suspended cord above-ground litter-trapping networks [4]. Two members of this group were sampled as part of this study, both likely representing previously undescribed species of *Marasmius*. BOR204 was a litter-trapping net suspended from the lower branches of a tree at head height, while BOR96 was in the form of a hyphal sheet on leaf litter (Figure 1b). Close matches to BOR204 (Appendix A)—MN930586 *Marasmius chrysocephalus* and MN930555 *Marasmius* sp.—were both sequences retrieved from a study of rhizomorph-producing fungi commonly associated with bird nests in Guyana [53].

Phallales include taxa known as stinkhorns and lattice stinkhorns, the latter being endemic to tropical regions. This order was represented by BOR101, which grouped closely to several *Phallus* species. It is thought that many, if not all of the Phallales originated from tropical regions as they are the most abundant and diverse in the tropics [54]. *Phallus* spp. have previously been identified as often forming white cords in forest leaf litter [14].

Taxonomic identification based on gross cord morphology alone was not possible, and hence the need for molecular identification. There were no hard boundaries between morphology types; however, cords were grouped by morphology as far as reasonably possible so that any potential patterns in the distribution of cord morphology across the phylogenetic trees could be linked to their molecular taxonomy. We did not find any obvious cord morphology patterns distinct to certain taxa or clades (Figure 3). It is possible that the different cord morphologies observed such as thickness are related to growth substrate or the age/life cycle stage of the fungi as opposed to a fixed morphology for that taxon [55]. This would explain the morphological differences observed between the collected fungi despite close genetic relationships. For example, BOR91 and BOR97 have identical sequences; however, they do not look identical (Figure 1m and Figure 2x, respectively); as such, they have been classified into different morphological groups. However, it is worth noting they are growing on different substrates, which could be an explanation for the morphological differences.

There are also differences between the collected CF fungi and certain close matches with fungi species in the database for which no evidence of cord formation is noted in the literature. It is possible that environmental factors present in tropical forest systems may promote the development of foraging cord networks in multiple fungal lineages. More comprehensive sampling could shed light on this unexplored area of fungal morphology.

The growth of basidiomycete mycelia is strongly influenced by the patchy distribution of organic substrates and mineral nutrients [2]. Therefore, it is not surprising that a wide diversity of fungi was seen at both sites, despite the geographical and ecological similarities of the two forest conservation areas. It is worth noting each CF fungal sample was collected because it looked slightly different as the purpose of collection in this study was also to assess the utility of fungal cord sampling as a rapid way of gaining insight into the diversity of active cord formers within the forest. It is possible that the corresponding fruiting bodies would have been found, but this would have taken significantly more time and effort. Furthermore even if the sporophores were present at the time, it would not necessarily enable the cord structures and fruiting bodies to be directly associated with each other.

The high proportions of novel sequence types detected in this study, even with a small sampling effort, may simply be due to the under-sampling of tropical Basidiomycetes, but may also represent the different perspective on biodiversity gained by sampling alternative functional structures. In any case, our results suggest that there is a large diversity of uncharacterised CF agaricomycete fungi at the two northern Borneo sites sampled.

All of the fungi collected in this study were previously unsequenced; however, it is worth noting that standard databases such as GenBank, UNITE, and SILVA do not fully represent the vast fungal diversity increasingly revealed by molecular studies, high-throughput sequencing in particular [56,57]. Not all known species are represented in sequence repositories, and thus (some of) the sequences generated in this study may represent described but unsequenced species, while the remainder represent novel, undescribed fungal diversity.

## 5. Conclusions

This study has revealed a diversity of tropical forest fungal sequences, none of which are represented in public databases, and may represent taxa new to science. Evolutionary radiations such as those exhibited by Clade A indicate a high diversity of niches for some fungal lineages in the sites sampled. In addition to surveying much more extensively than in the study, future research could investigate whether such radiations are specific to certain regions (endemism) or habitat types, or are more generally distributed in tropical forests. Sampling of corresponding fruiting bodies will enable further characterisation and descriptions of new taxa.

We emphasise the importance of continuing to explore previously unknown fungal diversity in tropical forests, having shown that even very small scale sampling can produce high proportions of new sequence types. We suggest that carbon and other nutrient cycling in such forests is significantly mediated by a largely unknown diversity of basidiomycete fungi. Studying fungal cord morphology is limited by the large number of undescribed species and subtle morphological characteristics that make correct identification difficult. However, sampling cords is relatively easy and yields a high diversity of fungi that are actively growing at the point of sampling. Molecular and phylogenetic techniques enable these taxa to be identified as far as current databases allow, as well as to reveal their mutual evolutionary relationships, and moreover those to other lineages for which phenotypic, ecological, and/or molecular data are available. They also provide the basis for group-specific molecular assays to be developed in order to investigate clades of interest in more targeted ways.

## Figures and Tables

**Figure 1 microorganisms-10-00239-f001:**
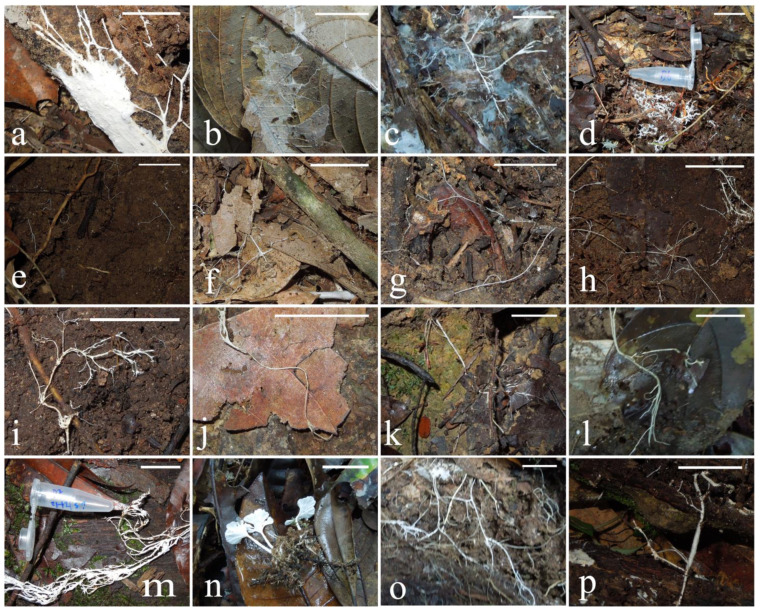
Photographs of fungi growing on leaf litter and small woody debris in situ prior to collection. Cords were loosely grouped according to thickness of cords. Scale bar is 2 cm. Sheet-forming: (**a**) BOR78, (**b**) BOR96, (**c**) BOR202, (**d**) BOR94. Thin cords: (**e**) BOR77, (**f**) BOR79, (**g**) BOR81, (**h**) BOR82. Thick cords: (**i**) BOR 83, (**j**) BOR86, (**k**) BOR88, (**l**) BOR99, (**m**) BOR91, (**n**) BOR110, (**o**) BOR201, (**p**) BOR80.

**Figure 2 microorganisms-10-00239-f002:**
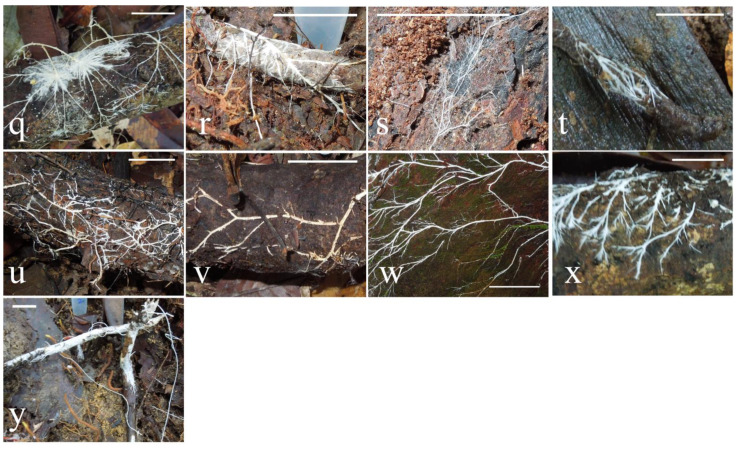
Photographs of fungi growing directly on tree branches in situ prior to collection. Scale bar is 2 cm. (**q**) BOR98, (**r**) BOR102, (**s**) BOR85, (**t**) BOR203, (**u**) BOR101, (**v**) BOR105, (**w**) BOR100, (**x**) BOR97, (**y**) BOR90.

**Figure 3 microorganisms-10-00239-f003:**
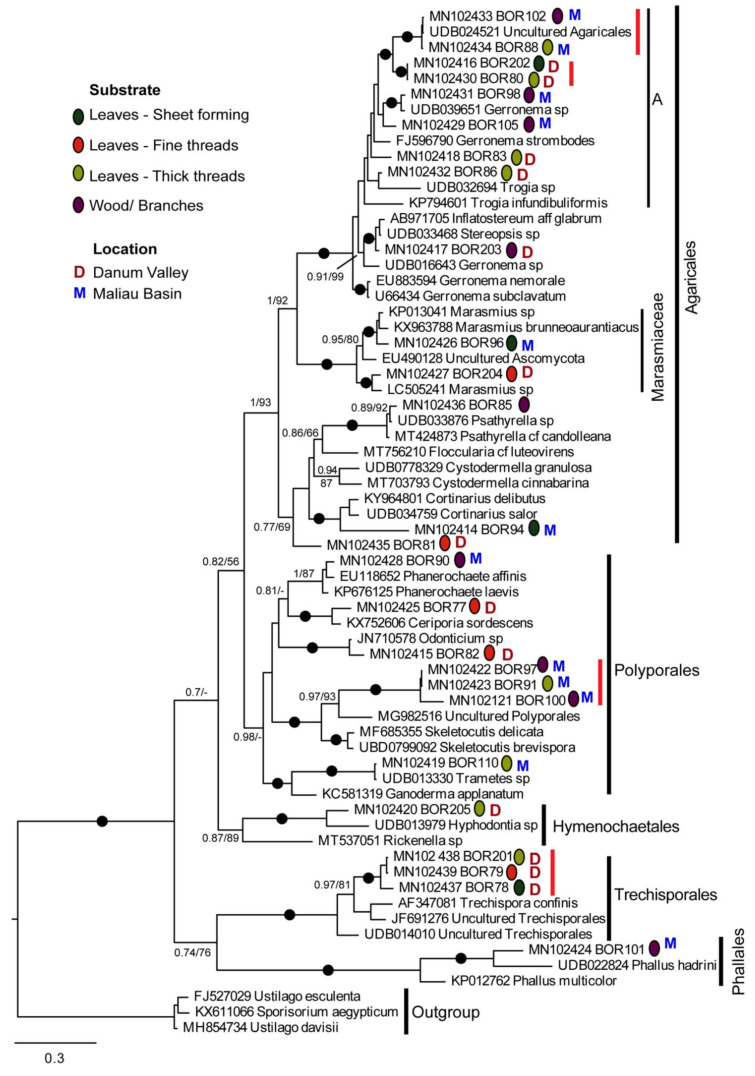
Bayesian phylogenetic tree of the ITS1-LR21 amplicon (3′ of 18S to 5′ of 28S rRNA genes). Sequences generated in this study are indicated by ‘BOR’. Vertical red lines indicate very similar BOR sequences that likely represent the same species. The analysis is intended to show the relationship of the BOR sequences to their closest relatives in GenBank and UNITE, as well as their ordinal affiliations, and is not appropriate for inferring higher level fungal relationships. Filled black circles indicate where both the Bayesian posterior probability (BPP) and maximum likelihood bootstrap (BS) support are above 0.95 and 95%, respectively. Other support values are given for nodes where BPP > 0.9 and/or BS > 75%. Supports are indicated in the order BPP/BS. ‘-’ indicates node not recovered by the method indicated.

## Data Availability

The sequence data presented in this study are deposited in in Genbank (accession numbers: MN102414-39).

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
