# Peer review of "Diversity and Phylogeny of Novel Cord-Forming Fungi from Borneo"

_microorganisms, 2022, doi:10.3390/microorganisms10020239_

Round 1

Reviewer 1 Report

Dear Authors,

I have now completed the review of your manuscript. Thank you for your effort, and congratulations on your work! It is a very interesting study, and I have only minor (although quite numerous) remarks and comments. I do believe that they will help you to improve your paper. Please find all my comments in the .docx file attached.

With best regards

Author Response

Dear Reviewer,

We thank you for taking the time to review our paper and for your helpful and informative feedback. I have attached a document with responses to all of your points. I hope you will find them to be satisfactory.

Kind regards,

Rachel Foster

Reviewer 2 Report

Reviewed paper deals with so far unexplored diversity of CF fungi. At first glance, the study is properly structured and could be read fluently. In general, I consider reviewed paper as a nice pivotal study of Bornean CF fungi, but I believe that this study could benefit from suggested improvements

1. You stated that you performed phylogenetic analysis in order to resolve the species pool of CF fungi in Borneo. Moreover, you concluded that your result showed that all analyzed specimens represent yet undescribed species. Though I am not doubting that result, I think that phylogenetic analysis was not conducted properly, lacks robustness and the results are not properly supported by the analysis. The main issues with your analysis are
a. It is not clear how you treat your dataset after alignment. How do you deal with hypervariable and hardly alignable regions of ITS? It is sometimes quite challenging to align ITS sequences throughout genus, not yet throughout such unrelated lineages. If you kept all the positions, it may in some cases result in long branch attractions etc.
b. it is not clear to me how you processed multiple sequences of the same species (such as BOR97, 100, 91). In the discussion, you mentioned that there were differences amongst them, obviously you have used only a single sequence to represent all of them. Did you use a consensus sequence? Also I do not think it is correct as those sequences should be blasted separately and all of them should be included in the tree
c. It is not clear what your sampling strategy is. You declared that you included the closest match retrieved from Blast search. However taxa included in the tree does not correspond with the declared strategy. For example, it is not clear why you chose Ganoderma applanatum as sister to Perenniporia ochroleuca, when there was comprehensive phylogeny of Perenniporia published and it is clear that P. ochroleuca is placed deep within the genus(see Zhao et al. 2013). Same cases could be found throughout analysis (e.g. Phanerochaete etc.)
d. Plenty of the terminal nodes are not supported by the analysis (e.g. BOR101, BOR85, BOR98 ...), therefore is is hard to say whether they are separate species or not

I would suggest to redo molecular analysis in this way
- include all of the sequences into analysis
- for each sequence include 3 closest BLAST hits, search also UNITE
- split analysis into several independent parts (e.g. Agaricales, Polyporales...), include more relevant taxa into the dataset (according to the recent published phylogenies) and run each analysis independently
- support your results also with Maximum Likelihood analysis

If you wish to keep everything together, than is better to analyze LSU region as used in Persoonia Fungal Planet Description Sheets

2. In the tree you kept collections numbers, they are not connected with GenBank numbers. Prepare at least table with collection numbers and GenBank numbers

3. Discussion is strongly focused on position of the species within the tree and together with lack of robust sampling, it may exhibit misleading statements (such as BOR90 is closely related to Phanerochaete from Sweden, I think this position is due to lack other Phanerochaete species in the analysis)
I am missing summary of Borneo´s fungal diversity and comparison of already published species which may be potentially cord forming (f.e. There are 2 species of Perenniporia described from Malaysia by Hattori & Lee 1999). Also, I miss the part about comparison of Tropic and Neotropic CF diversity.

Few other remarks
Line 201, 69, 214 - improve references

Line 84: authors declare they want to gain more insight into CF fungi phylogeny. However, in this term, phylogeny is not conducted properly. To resolve phylogenetic relationships, more robust sampling within lineages is necessary. Maybe it is better to use word evolution

Line 112: LR21 is reverse primer, while ITS1 is forward (see https://unite.ut.ee/primers.php)

Line 124: Specify, how authors treat hyper variable regions of ITS in alignment? Did they use gblocks?

Line 223: You declared, that you have used assembled sequences, therefore sequencing error should be fixed

Figure 3: Tree should be provided in higher resolution

References need in-depth revision 

Author Response

Dear Reviewer,

We thank you for taking the time to review our paper and for your helpful and informative feedback. I have attached a document with responses in red text to all of your points. I hope you will find them to be satisfactory.

Kind regards,

Rachel Foster

Round 2

Reviewer 2 Report

Authors met all my objections and suggestions. I am gladly recommending reviewed manuscript for publication.

My only comment would be, that I still recommend thorough reading during proofs and remove small typos